# A Framework for Identifying the Critical Region in Water Distribution Network for Reinforcement Strategy from Preparation Resilience

**Mingyuan Zhang [1],\*** , **Juan Zhang [1]** , **Gang Li [2] and Yuan Zhao [1]**

[1] Department of Construction Management, Dalian University of Technology, Dalian 116000, China; zhangjuandlut@mail.dlut.edu.cn (J.Z.); zy0909@mail.dlut.edu.cn (Y.Z.)

[2] School of Civil Engineering, Dalian University of Technology, Dalian 116000, China; gli@dlut.edu.cn

\* Correspondence: myzhang@dlut.edu.cn

**Abstract:** Water distribution networks (WDNs), an interconnected collection of hydraulic control elements, are susceptible to a small disturbance that may induce unbalancing flows within a WDN and trigger large-scale losses and secondary failures. Identifying critical regions in a water distribution network (WDN) to formulate a scientific reinforcement strategy is significant for improving the resilience when network disruption occurs. This paper proposes a framework that identifies critical regions within WDNs, based on the three metrics that integrate the characteristics of WDNs with an external service function; the criticality of urban function zones, nodal supply water level and water shortage. Then, the identified critical regions are reinforced to minimize service loss due to disruptions. The framework was applied for a WDN in Dalian, China, as a case study. The results showed the framework efficiently identified critical regions required for effective WDN reinforcements. In addition, this study shows that the attributes of urban function zones play an important role in the distribution of water shortage and service loss of each region.

**Keywords:** water distribution network; cascading failures; infrastructure disruption; infrastructure resilience; critical region

---

## 1. Introduction

The functioning and resilience of modern societies have become more and more dependent on critical infrastructures. As one of the most critical infrastructures, the water distribution network (WDN) plays an important role in the normal operation of modern societies. However, disruptive events, such as intentional attacks, natural disasters or human-caused accidents (e.g., destructive earthquakes and terrorist attacks), can have significant consequences on WDNs composed of numerous interconnected functional and structural components [1,2]. The interconnections can improve the efficiency of the WDNs, but small disturbances trigger cascading failures, causing large-scale consequences [3,4]. The destruction of WDN causes the water shortage, which directly interferes with the normal operation of society and impacts people's life. Hence, the WDS security issues are worthy of detailed deliberation to build more resilient WDNs.

To reduce the impact of disruptive events on infrastructures, the concept of resilience has been widely proposed in the field of infrastructure research [5–9]. Despite the differences in the details of resilience concept, resilience has been generally defined as the capacity of a system to resist (preparation phase), absorb and withstand (responding phase), and rapidly recover from (restoration phase) exceptional conditions [10–12], it reflects the capacity of the critical infrastructures to maintain service function before, during and after disruptions from a more comprehensive perspective. The present

study focused on reinforcement strategies (i.e., improving reliability/robustness) that can reduce damages owing to disruptive events, using an actual WDN as an example. The level of resilience is the same as spring extension, which can only manifest under interferences, and the stress-strain analysis in mechanics can be used to evaluate the resilience level of the infrastructure systems, that is to evaluate the performance of the systems (i.e., strain) under different levels of disturbances (i.e., stress) [13,14]. This kind of performance-based method avoids directly describing all the possible risks that could cause damage to the system; it is quite difficult, if not impossible. Instead, the failure modes of infrastructure systems are relatively easy to identify and analyze [15–18]. Therefore, the level of systems malfunction is commonly modeled by two failure modes, namely, node failures and pipe breaks, rather than describing the possible threats. The two failure modes of node failures and pipe breaks are considered in this paper.

In general, it is not economic to reinforce all components of the critical infrastructure systems due to the limitation of reinforcement resources, so measuring the criticality and reliability of the components to identify the critical region in the infrastructure systems to provide effective guidance for reinforcement strategies is of great significance. In existing studies, one part measured the criticality of components with the various centrality measures from a structure viewpoint. The classical topological centrality measures include the degree centrality [19], the closeness centrality [20], and the betweenness centrality [21]. However, few studies have used the nodal supply water level as a metric to measure the structural importance of the components in the infrastructure systems. Additionally, the other part measured the criticality or reliability of components with flow analysis from a function viewpoint. The component's critical measures based on cascading failure progress were analyzed by Enrico and Giovanni [22]. Shuang et al. evaluated the vulnerability of nodes under cascading failures [4]. Shuang et al. also identified the crucial pipes by introducing hydraulic analysis and cascading failures [2]. Zainab et al. presented a flow-based model to identify the optimal set of pipeline replacement strategies [23]. As one of the critical infrastructure systems in urban landscapes, WDN is a network composed of numerous interconnected and interacting components, and it associates with the urban function zones through supplying water to urban areas. In general, the same water shortage causes greater losses to the urban function zones, which are more necessary and critical to people's life (e.g., hospitals) [24]. Nevertheless, none of the existing component's importance measures take into account the criticality of urban function zones, which is an important factor for evaluating losses caused by the disruption of components. In order to comprehensively assess the losses to the urban caused by the malfunction of the WDN, not just from the viewpoint of the performance degradation of the WDN itself, this paper takes the criticality of urban function zones into components importance measures. Furthermore, the critical region where important components are distributed can be identified. The region criticality assessment can provide the effective guidance for the reinforcement strategy of WDNs, especially within the limited reinforcement resources.

On the basis of the above idea, this paper proposes a methodology framework that identifies critical regions within WDNs to formulate reinforcement strategies for improving preparation resilience. The framework takes into account the distribution and criticality of urban function zones, the structural attributes and the hydraulic function of WDNs.

This paper is organized as follows. The resilience definition and hydraulic calculation for WDN under a balanced state are provided in Section 2.1. The failure modeling and pressure-driven hydraulic simulation under a WDN disruption are introduced in Section 2.2. The framework is introduced in Section 2.3. An application of the framework to an actual WDN, including the identification of the critical regions and effective reinforcement strategies for the WDN, is provided in Section 3. The concluding remarks are drawn in Section 4.

## 2. Materials and Methods

### 2.1. System Resilience Definition and Hydraulic Calculation for WDN

#### 2.1.1. System Resilience

A resilience framework of an infrastructure system going from disruption to normal state is depicted as shown in Figure 1. In the framework of system resilience, the reduction of damage is divided into two stages. The first stage (0-$t_r$) is to reinforce the infrastructure to minimize the loss of system functions from component damages driven by disruptive events. The second stage ($t_r$-$t_T$) is to efficiently allocate repair resources to promptly recover the system from the damages. From Figure 1, the infrastructure system operates at a normal value $F_0$. until suffering disruptive events at time $t_e$. Suppose that the disruptions on the infrastructure system deteriorate its performance to $F_{min}$, and the infrastructure system starts to recover after emergency preparation at time $t_r$, ending at a level the same, close to, or better than the original performance level $F_0$ at time $t_T$. The severity of the disruptive events can be measured by (1) the decline in infrastructure system performance in the first stage, and (2) the infrastructure system recovery time and trajectory in the second stage. As the time of emergency preparation is quite shorter than the recovery period, the total damages from the disruption events can be reduced by minimizing the performance decline in the first stage and optimizing the recovery time and trajectory. The overall resilience process during the first stage can be enhanced by proactive plans such as predictable events. In addition, more advanced recovery techniques are required during the second phase for rapid and effective infrastructure system recovery. In this paper, the critical regions of the system are identified through failure simulation, to strengthen the critical regions in the normal state, and then minimize the loss of the system in the first stage.

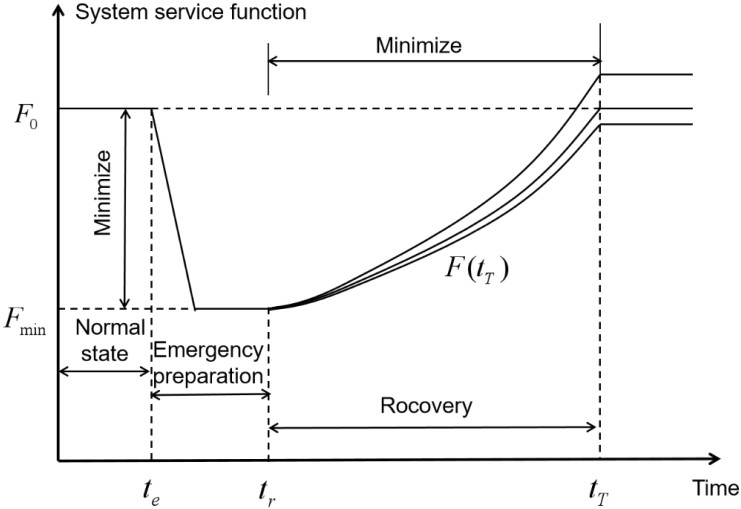

**Figure 1.** The resilience framework of infrastructure system.

#### 2.1.2. Representation and Hydraulic Calculation for WDNs

A WDN is usually modelled by a network that is comprised of points inter-connected by links with a set of attributes and information on physical components [13]. The primary components of WDN are nodes and pipes; nodes are presented by points are defined by nodal demand and head, while pipes denoted by links with lengths, diameters, flow direction and other physical attributes. A WDN is usually formulated by a directed graph that indicates an operational flow and nodal pressure. The directed graph can be expressed as G (*V, E*), where *V* and *E* are the set of nodes and pipes, respectively. Furthermore, an essential base for the analysis of a WDN topological structure is an incidence matrix that represents the structure of the graph by its elements.

In the incidence matrix $[A]$, the rows and columns of the matrix correspond to the nodes and pipes of the WDN, respectively, and its element $a_{ij}$ is assumed as follows [25]:

$$a_{ij} = \begin{cases} 1, \text{node } i \text{ is the end node of pipe } j \\ -1, \text{node } i \text{ is the starting node of pipe } j \\ 0, \text{pipe } j \text{ is not connected with } i \end{cases}, \tag{1}$$

Under disruptive events in WDNs, some components (e.g., nodes and pipes) may be damaged while other operational nodes in the WDNs follow the laws of conservation of mass and energy. The node flow equation and node pressure equation are adopted to determine flow distributions and pressure throughout the WDNs, respectively.

For an operational node, the formulation is as follows:

$$\sum Q_{in} - \sum Q_{out} = q_{ext}, \tag{2}$$

where $Q_{in}$ and $Q_{out}$ are inflow and outflow at the node, respectively, and $q_{ext}$ is the external supply or demand.

A general form for any path is as follows:

$$\sum_{u \in U_p} h_{j,u} + \sum_{v \in V_p} h_{p,v} = \Delta E, \tag{3}$$

where $h_{j,u}$ is the head-loss across component $u$ along the path, $h_{P,v}$ is the head added by pump $v$ and $\Delta E$ is the difference in head between the end nodes on a path. The friction in pipes ($h_j$) is calculated by the Hazen–Williams equation:

$$h_j = KQ^{1.852} = 10.654 * \left(\frac{Q}{C}\right)^{1.852} * \frac{1}{D^{4.87}} * L, \tag{4}$$

where $K$ is the pipe coefficient which depends on the diameter of pipe $D$, the length of pipe $L$, the material of the pipe and $C$ is the Hazen–Williams coefficient.

For the closed loops, the formulation is as follows:

$$\sum_{j \in J_L} K_j Q_j |Q_j|^{0.852} = 0, \tag{5}$$

where $J_L$ is the set of pipes in a certain loop.

### 2.2. Failure Mode and Pressure-Driven Analysis

### 2.2.1. The Spatially Localized Attacks and Cascading Failures

Based on the existing studies about the vulnerability and reliability of infrastructure systems, the failure situations are grouped into three types, random failures, malevolent attacks and spatial failures [26–28]. The Spatially Localized Attack-induced impact on the infrastructure systems is modeled as the failure of components that located in close proximity in a localized area while those outside this area remain operating [28]. Based on the characteristics of WDN and the principle of geographical close proximity, the area of the WDN can be divided into multiple regions where if a region is subjected to damage, all the components distributed in close proximity to each other within that region will be completely cut off, and hence removed from the WDN.

### 2.2.2. Pressure-Driven Analysis

Mechanical failure in a WDN causes insufficient pressure, which induces a water supply failure. This study conducted a pressure-driven analysis, a cutting-edge hydraulic simulation to identify how a WDN performs under insufficient pressure conditions. Compared to demand-driven modeling methods, the pressure-driven analysis provides a more hydraulically consistent behavior of a network. Furthermore, the pressure-driven simulation can prevent the negative pressure on nodes, which is more realistic. Among various formulas to simulate the pressure-discharge relationship, many researchers have actively employed a formula proposed by Wagner [29] to detect component failures [30–33] in a system, which is expressed as follows:

$$Q_k^{act} = \begin{cases} 0, & P_k \leq P_k^{min} \\ Q_k^{req} \left( \frac{P_k - P_k^{min}}{P_k^{ser} - P_k^{min}} \right)^{\frac{1}{n}}, & P_k^{min} < P_k < P_k^{ser}, \\ Q_k^{req}, & P_k^{ser} \leq P_k \end{cases} \tag{6}$$

where $n$ is the pressure exponent which is usually taken as 1.5~2 ( $n$ is set to 2 in this study), $Q_k^{act}$ is the actual demand at node $k$, $Q_k^{req}$ is the required demand at node $k$, $P_k^{ser}$ is the service pressure to supply the required demand at node $k$, $P_k$ is the available pressure at node $k$ and $P_k^{min}$ is the minimum pressure to initiate water supply at node $k$.

A WDN under a failure, e.g., the closure of damaged pipes and nodes, interrupts the water supply to users. Therefore, the actual demand needs to be modified as follows before hydraulic simulation:

$$Q_k^{req} = Q_{k,0}^{req} \frac{n_k}{n_{k,o}}, \tag{7}$$

where $n_k$ and $n_{k,0}$ are the number of pipes still connected to node $k$ in the case of a node or pipe failure and those connected at the normal state, respectively, and $Q_{k,0}^{req}$ denotes the nodal demand at the normal state.

### 2.3. A Framework for Identifying the Critical Region in WDN

#### 2.3.1. Evaluating the Weight of Urban Functional Zones

Note that the same water shortage cause the greater losses to more important function zones (e.g., hospitals) than less important entertainment zones (e.g., parks); the importance of the function zones supplied by nodes in a WDN is considered to quantify such a distinction. Under the process of urbanization accelerated along with economic development, a community is gradually developed into a complex composed of a variety of function zones (the buildings supporting each of these essential urban functions are referred herein as urban function sector). In general, an urban area mainly consists of five basic function zones: residential zones, business zones, medical zones, education zones, and public service and entertainment zones [34,35]. Among the five basic function zones, the function zones in charge of critical services (e.g., hospitals) are more important than other zones under a water shortage status. For the same kind of function sector, the greater the service capacity, the greater its importance. Therefore, the importance evaluation for the urban function zones can be divided into two parts: (1) inter-type (among the five basic function zones) and (2) intra-type (among each type of five basic function zones).

In this study, the inter-type relative importance for the urban function zones was determined by the Maslow's hierarchy of needs [35]. As Maslow believes that human needs are divided into basic needs and advanced needs, and the needs are hierarchical, that is, when a person meets the basic needs, he/she will generally pursue advanced needs. Human needs generally appear in the order of physiological needs, safety needs, belongingness and love needs, esteem needs and self-actualization needs. The Maslow's hierarchy of needs is shown in Figure 2.

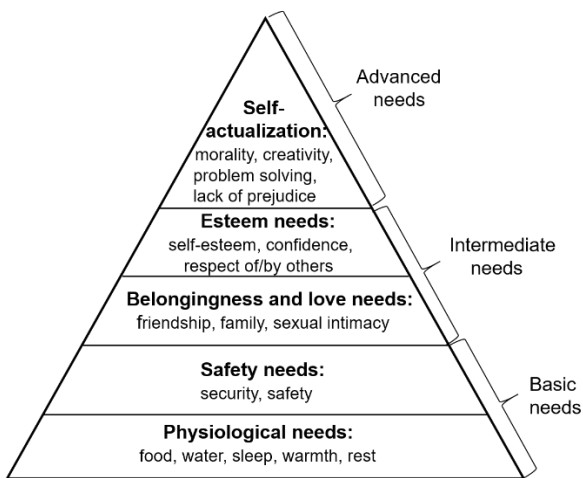

**Figure 2.** Maslow's hierarchy of needs.

Based on the Maslow's hierarchy of needs, the urban function zones are grouped into two types: necessary and optional. The necessary type of function zones provides the basic needs such as housing, food and safety to humans, corresponding to residential zones, business zones and medical zones in this paper. The optional type of function zones provides human beings with intimate relationships, self-fulfillment and other advanced needs, corresponding to education zones and public service and entertainment zones in this paper. The inter-type relative importance of function zones varies with different countries, cultures and religions, and is subjective, so it is quite difficult to calculate quantitatively; in this paper, it is simply assumed that the relative importance for the necessary function zones is twice that of the optional function zones.

As the same kind of function zone, the greater the service capacity, the greater its importance. In the medical zones, hospitals are categorized according to the service capacity measured by the number of beds, personnel, equipment and floor space, based on which we then introduce the number of beds in the hospital to measure the service capacity of a hospital. The service capacity of other function zones is also measured in this way. The service capacity for the residential community and education zones is measured by the number of households and teachers, respectively. For the business zones, public service and recreation zones, the service capacity is measured by the areas corresponding to each function. The formula to calculate the intra-type relative weight of the function zones is as follows:

$$w_{i,j} = \frac{S_{i,j} - \min\{S_i\}}{\max\{S_i\} - \min\{S_i\}} \tag{8}$$

where $i$ is the type of function zones, $i = 1, 2, 3, 4, 5$, corresponding to the residential zones, business zones, medical zones, education zones and public service and entertainment zones, respectively. $j$ is a residential community, a shopping mall, a hospital, a school, governmental agencies or parks. $S$ is a set of the indicator values for the five basic function zones.

The information on which urban function zones supplied by nodes in a WDN is not available for analysis due to security and privacy concerns. Thus, this paper assumed that each urban function sector supplies services to a node closest to the zones in geographical proximity.

2.3.2. Evaluating the Structural and Functional Importance of Regions

A concept of the nodal water supply level was introduced to measure the structural importance (SI) of nodes from a topological structure point [36]. In WDNs, the water supply level for a node directly connected to the water source node is defined as level 1 and the water supply level for other nodes is determined step-by-step according to the flow direction. When the water supply level for a node on different lines is inconsistent, a higher water supply level was assigned to the node as the water supply level. In Figure 3, the water supply levels (WSL) of nodes are shown in Table 1. The closer

the node is to the water source, the higher the water supply level of the node is, and the greater the impact of its destruction on the entire WDN. Therefore, the nodal water supply level can effectively reflect the structural importance of nodes in a WDN. The structural importance of region $R$ ($SI_R$) is:

$$SI_R = \frac{1}{N} \sum_{k=1}^{N} WSL_k \tag{9}$$

in which $WSL_k$ is the water supply level of node $k$ in region $R$, and $N$ is the number of nodes in region $R$.

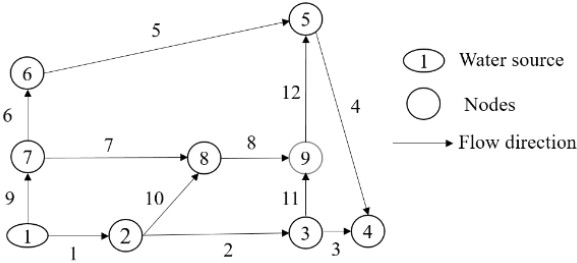

**Figure 3.** The line diagram of the water distribution network.

**Table 1.** The water supply level of nodes in Figure 3.

| Node # | WSL | Node # | WSL |
|--------|-----|--------|-----|
| 2 | 1 | 6 | 2 |
| 3 | 2 | 7 | 1 |
| 4 | 3 | 8 | 2 |
| 5 | 3 | 9 | 3 |

Given some nodes and pipes in a failure status within a damaged WDN, the water flow in the WDN may be redistributed corresponding to the interconnected and interacting components, resulting in deficient pressure at some nodes and subsequently dissatisfying the normal demand. A larger water shortage through the entire WDN induced by a damaged node indicates that the node plays a more important role in the WDN. In other words, the amount of water shortage of the WDN induced by a failed node can reflect the relative importance of the node. In addition to evaluating the topological structure importance of the water supply network nodes, therefore, the functional importance (FI) of each node was also measured by the amount of water shortage induced by each node. The water shortage (WS) of node $k$ can be calculated using Equation (10):

$$WS_k = 1 - \frac{Q_k^{arc}}{Q_k^{req}} \tag{10}$$

The water shortage of region $r$ ($WS_r^{\text{initial}}$) is:

$$WS_r^{\text{initial}} = \frac{1}{N} \sum_{k=1}^{N} WS_k \tag{11}$$

The mean water shortage of other regions due to the disruption of region $R$:

$$M\left(WS_r^{\text{initial}}\right)_R = \frac{1}{M} \sum_{r=1, r \neq R}^{M} WS_r^{\text{initial}} \tag{12}$$

The water shortage considering function zone importance of region $r$ ($WS_r$) is:

$$WS_r = \frac{1}{N} \sum_{k=1}^{N} w_{i,j}^{k} \cdot WS_k \tag{13}$$

The functional importance (FI) of region $R$ can be measured by calculating the mean water shortage of other regions due to the disruption of region $R$:

$$FI_R = \frac{1}{M} \sum_{r=1,r\neq R}^{M} WS_r \tag{14}$$

in which, $M$ is the number of regions in WDN.

### 2.3.3. Identifying the Critical Regions

As one of the most critical infrastructures, a WDN supplies the water demands to the urban area through numerous components (e.g., nodes and pipes) complicatedly interconnected each other and the small disturbance can trigger large-scale consequences and secondary failures. This severe situation is defined as cascading failures. Once a network encounters spatial attacks, the components located in the localized area are damaged, which may trigger the cascading failures that induce the degradation of WDN performance. This study adopted spatial attacks to identify critical regions under the cascading failures. The regions (including some nodes and pipes) in WDN will be damaged due to disasters, further resulting in water shortage. In general, the same water shortage causes greater losses to the urban function zones which are more critical to people's life, such as hospitals and residences, but fewer losses to entertainment zones. So, in practice, the most critical region in the WDN is not only that its failure has a great impact on the WDN itself, but also that its failure has a great impact on urban water supplying. The critical regions depend on the topological region within the WDN, the water supply capacity and the criticality of the urban function zones served. Given limited reinforcement resources, it is necessary to formulate a scientific pre-disaster reinforcement plan with a priority on more critical regions in the WDN. Therefore, it is a prerequisite to identify critical regions in the WDN. This study proposed a systematic approach that consists of three steps to comprehensively identify the critical region in the WDN consists of three steps: (1) determining the water supply level of all nodes and the SI of regions; (2) analyzing the region FI considering the distribution of urban function zones and the water shortage; and (3) identifying the critical region by integrating the previous two steps. The critical importance (CI) of region $R$ is:

$$CI_R = \alpha \frac{\max\{\boldsymbol{SI}\} - SI_R}{\max\{\boldsymbol{SI}\} - \min\{\boldsymbol{SI}\}} + \beta \frac{FI_R - \min\{\boldsymbol{FI}\}}{\max\{\boldsymbol{FI}\} - \min\{\boldsymbol{FI}\}} \tag{15}$$

where $CI_R$ is the critical importance value of region $R$. The region $R$ with high $CI_R$ indicates that the region is more critical. $\boldsymbol{SI}$ is the set of structural importance of all regions in the WSD. $\boldsymbol{FI}$ is the set of functional importance of all regions in the WSD. $\alpha$ and $\beta$ are the structural importance coefficient and the functional importance coefficient, respectively, and $\alpha + \beta = 1$. The value of $\alpha$ and $\beta$ can be determined through expert judgment. In the paper, $\alpha = \beta = 0.5$. It is assumed that nodes and pipes in the WDN are either operational or a complete failure; the failure nodes and their connected pipes were removed from the rest of the WDN. The procedure of identifying the critical regions in the WDN is illustrated in Figure 4.

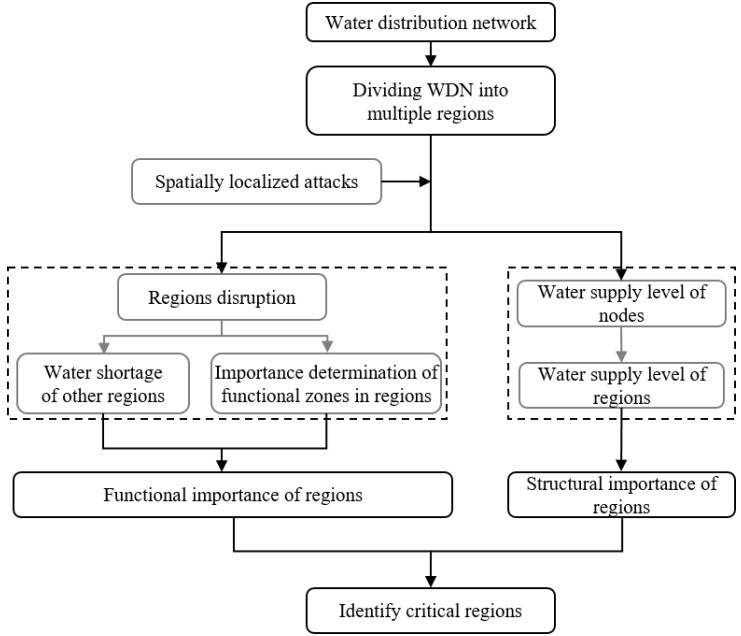

**Figure 4.** The procedure of identifying the critical regions in the water distribution network (WDN).

## 3. Results

The framework suggested in this study was applied to a WDN in Dalian, China, as shown as in Figure 5. The WDN consists of a single water source node, seventy-four demand nodes and ninety-four pipes. The WDN was divided into twenty regions based on the information of the WDN and geographical proximity. Parameters of nodes include required demand and elevation, and pipes include flow direction, diameter and length. Table 2 shows the parameters of nodes and pipes (The pipe number is represented by the nodes at both ends of the pipe, such as pipe 2–3 and pipe 27–28). The Hazen–William coefficient is 130. The minimum pressure is 28 m to ensure the basic operation.

**Table 2.** Parameters of WDN.

| Node# | Region# | Required Demand(L/s) | Elevation(m) | Pipe# | Region# | Diameter (mm) | Length (m) |
|---|---|---|---|---|---|---|---|
| 2 | 1 | 0.83 | 16.01 | 2–3 | 1 | 300 | 150.90 |
| 27 | 2 | 0 | 16.56 | 27–28 | 2 | 200 | 312.60 |
| 5 | 3 | 30.75 | 18.11 | 5–6 | 3 | 200 | 347.80 |
| 24 | 4 | 28.80 | 18.80 | 24–25 | 4 | 300 | 406.90 |
| 48 | 5 | 56.98 | 12.60 | 48–49 | 5 | 1000 | 265.10 |
| 11 | 6 | 19.04 | 20.64 | 11–12 | 6 | 500 | 556.70 |
| 45 | 7 | 20.53 | 24.60 | 45–46 | 7 | 200 | 584.40 |
| 23 | 8 | 23.70 | 32.33 | 23–17 | 8 | 200 | 329.20 |
| 18 | 9 | 42.05 | 24.60 | 18–19 | 9 | 300 | 424.20 |
| 1 | 10 | 21.44 | 16.14 | 1–26 | 10 | 700 | 367.20 |
| 29 | 11 | 16.27 | 21.35 | 29–30 | 11 | 200 | 217.30 |
| 32 | 12 | 11.94 | 19.42 | 32–33 | 12 | 200 | 279.80 |
| 38 | 13 | 1.13 | 21.30 | 38–39 | 13 | 200 | 277.80 |
| 43 | 14 | 20.53 | 11.00 | 43–44 | 14 | 300 | 194.60 |
| 52 | 15 | 28.84 | 27.72 | 52–53 | 15 | 1000 | 467.70 |
| 54 | 16 | 69.79 | 29.25 | 54–55 | 16 | 1000 | 410.10 |
| 55 | 17 | 59.10 | 32.04 | 55–66 | 17 | 700 | 488.10 |
| 60 | 18 | 12.65 | 66.40 | 60–61 | 18 | 200 | 409.60 |
| 65 | 19 | 56.98 | 25.00 | 65–66 | 19 | 500 | 457.20 |
| 68 | 20 | 33.60 | 38.25 | 68–69 | 20 | 300 | 532.10 |

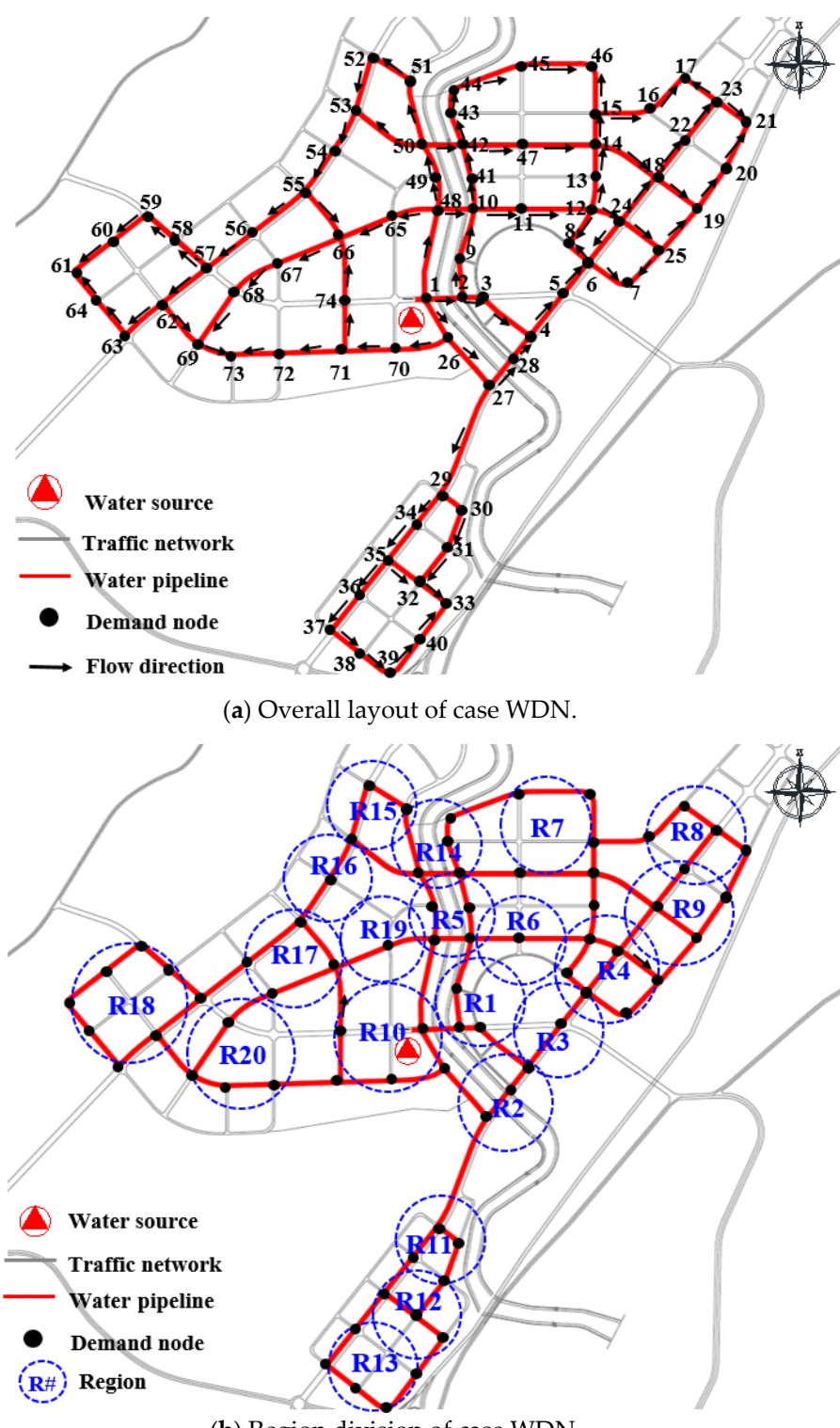

(**a**) Overall layout of case WDN.

(**b**) Region division of case WDN

**Figure 5.** Overall layout and region division of case WDN.

Table 3 shows the water supply level of regions in the WDN. The water supply level of the region 1 was the highest (the smallest level number is the highest water supply level) while the lowest was for region 18, indicating that region 1 is the closest to the water source and region 18 is the farthest based on the number of pipes and flow direction in the WDN. In other words, a disruption in region 1 may cause the largest impact on the entire WDN from a topological structure point while region 18 has

relatively fewer impacts. Therefore, the nodal water supply level metric is an effective indicator that reflects the structural importance of each region. In the case WDN, the structural importance of region 1 is the largest, while the smallest is for region 18.

**Table 3.** Water supply level of each region.

| Region #. | 1 | 2 | 3 | 4 | 5 | 6 | 7 | 8 | 9 | 10 |
|---|---|---|---|---|---|---|---|---|---|---|
| Water supply level | 2.67 | 3.67 | 5.00 | 6.14 | 3.00 | 4.00 | 7.60 | 9.50 | 8.00 | 3.50 |
| Region #. | 11 | 12 | 13 | 14 | 15 | 16 | 17 | 18 | 19 | 20 |
| Water supply level | 5.00 | 7.00 | 9.00 | 5.50 | 5.33 | 6.00 | 6.00 | 11.00 | 3.00 | 6.00 |

Under disruptions in a region, the components (i.e., nodes and pipes) within the region may be in a failure status, leading to ruining equilibrium and subsequently redistributing the water flow throughout the WDN. This condition results in deficient pressure for some nodes and consequently insufficient water supply with regard to the actual demands in some regions. The water shortages (the importance of functional zones is not considered) for regions in the WDN under a region (region 2/5/6/10/14/20) failure are shown in Figure 6. $WS^{initial}$ is the water shortage (The importance of functional zone is not considered), $M\left(WS^{initial}\right)$ is the mean water shortage (The importance of functional zone is not considered) of other regions due to the disruption of a region.

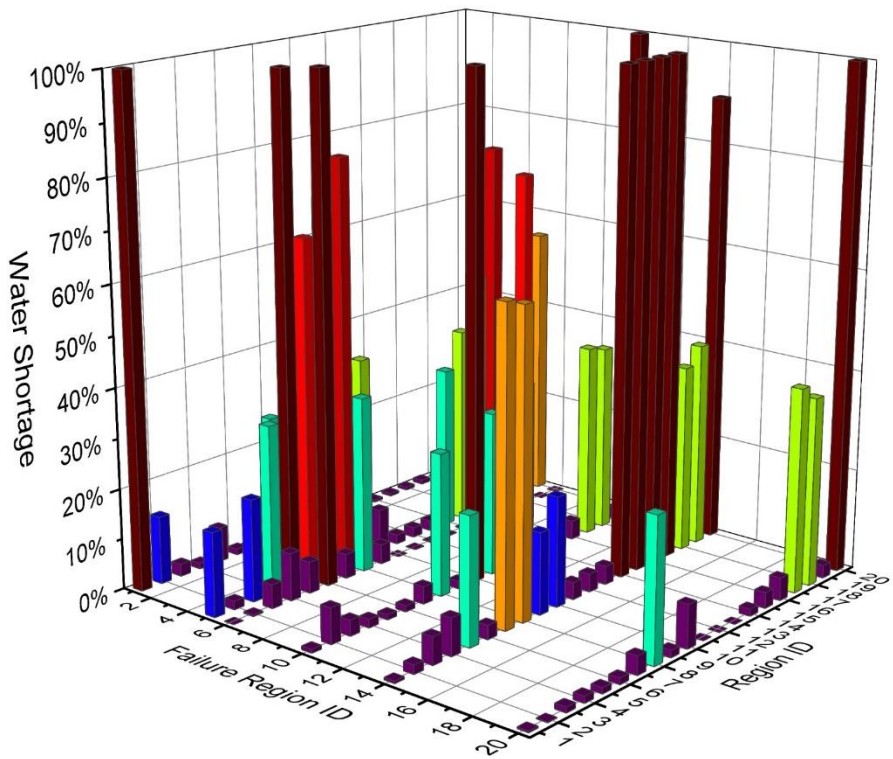

**Figure 6.** The $WS^{initial}$ of regions (Region ID) under a region failure (Failure Region ID).

The results show that the water shortage in a failure area was 100% because all nodes and pipes in this region were in a failure mode with regard to their water supply function. Moreover, the disruption in region 10 completely caused the water shortage in region 20, and the disruptions of region 14 completely deprived the water supply function for regions 14, 15, 16 and 17, i.e., the water shortage is 100%. The disruption of region 14 results in 100% water shortage in four regions, while the disruption in region 5 only causes 100% water shortage in one region. However, the disruption of region 5 leads to a water shortage of more than 40% in most regions. Therefore, it is not obvious to get the critical

regions from Figure 6. Hence, the comprehensive mean value of water shortage in all other regions caused by a region disruption is further used to reflect the critical region. Figure 7 shows the mean water shortages (the importance of functional zone is not considered) that occurred in other regions under the destruction of each region is section may be divided by subheadings. The results show that the damage in regions 5 and 14 caused a considerable mean water shortage in the WDN, of 40.49% and 39.17%, respectively, while the damage in regions 3, 18 and 19 induce a relatively smaller mean water shortage in the WDN of 5.73%, 6.04% and 5.63%, respectively. This result indicates that regions 5 and 14 play a key role in supplying water demands in the WDN, i.e., the water supply importance for regions 5 and 14 are larger while the contribution of regions 3, 18 and 19 to the water supply is relatively smaller. The results of water supply importance obtained by mean water shortage analysis are similar to the results of regional structure importance analysis mentioned above, but there are also differences. For example, both of them show that region 18 is relatively unimportant, but region 1 is the most important area in structural importance, while region 5 and region 14 are the most important in water supply importance. Thus, it is necessary to use the critical importance (CI) parameter proposed in Section 2.3.1 to synthesize the two parameters (the SI and FI) to evaluate the regional importance more comprehensively.

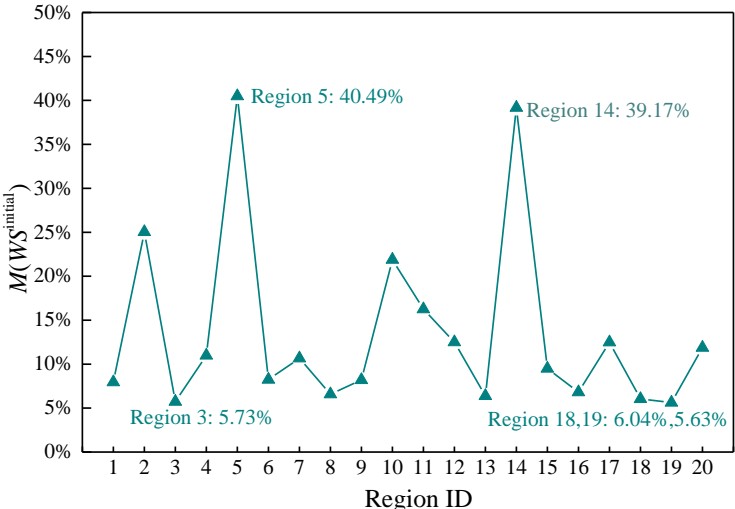

**Figure 7.** $M(WS^{\text{initial}})$ in WDN under a region disruption.

In addition, the same water shortage leads to a greater loss to the urban function zones in charge of critical service (e.g., hospitals), and it is not accurate to evaluate the loss of only by the water shortage in each region. Hence, the distribution characteristics of function zones are considered, and the water shortage considering the function zone importance of a region (*WS*) is used to indicate the region loss. Figure 8 shows the general distribution of function zones supplied in the WDN. Considering the water shortage and the distribution characteristics of urban function zones, the water shortage by a disruption in the WDN can be analyzed comprehensively.

Figure 9 shows a comparative analysis of mean water shortage (the importance of a functional zone is not considered) of other regions due to the disruption of a region ( $M(WS^{initial})$) and functional importance (*FI*). In Figure 9, the *FI* values of region 5 and region 14 are 100% and 94.95%, respectively. It is relatively larger than $M(WS^{\text{initial}})$. The reason of this is that region 5 has hospital functional zone (in Figure 8) and region 5 will lead to water shortage of more than 40% in most regions (in Figure 6); Region 14 has not hospital functional zone, and it will lead to water shortage of 100% in most regions (in Figure 6). Moreover, because there are hospital functional zones in regions 4, 5, 7, 15, 17 and 20 (in Figure 8), the *FI* values of these regions are relatively larger than $M(WS^{\text{initial}})$ values. There is a water source in region 10, so the *FI* value of region 10 is relatively large. The *FI* values of region 11, 12 and 13 are smaller than the value of $M(WS^{\text{initial}})$. The *FI* values and $M(WS^{\text{initial}})$ values of other

regions are basically equal. According to the *FI* value in Figure 9, region 5 and region 14 are of great functional importance. Therefore, in order to further clarify the impact of the disruption of region 5 and region 14 on other regions, the water shortages caused by the disruption of region 5 and region 14 considering the importance of functional zones and without considering the importance of functional zones are analyzed in detail.

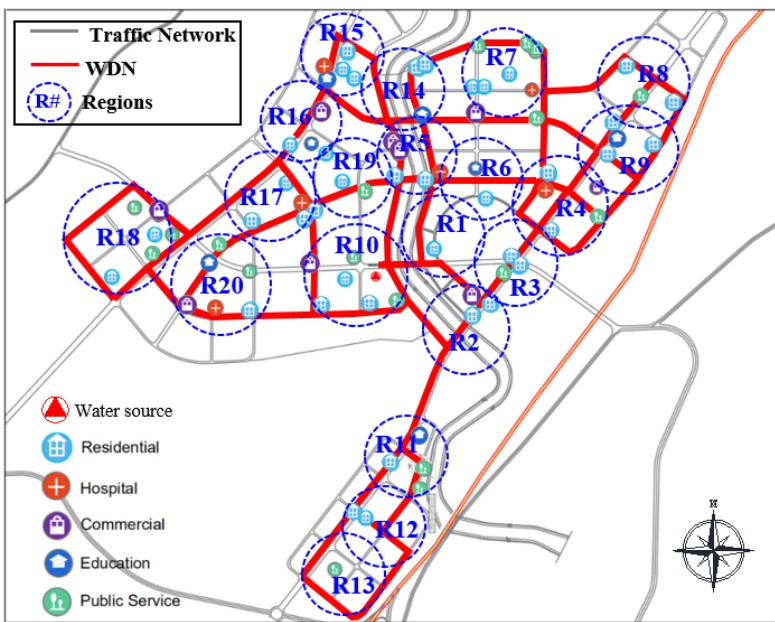

**Figure 8.** The distribution of urban function zones supplied by case WDN.

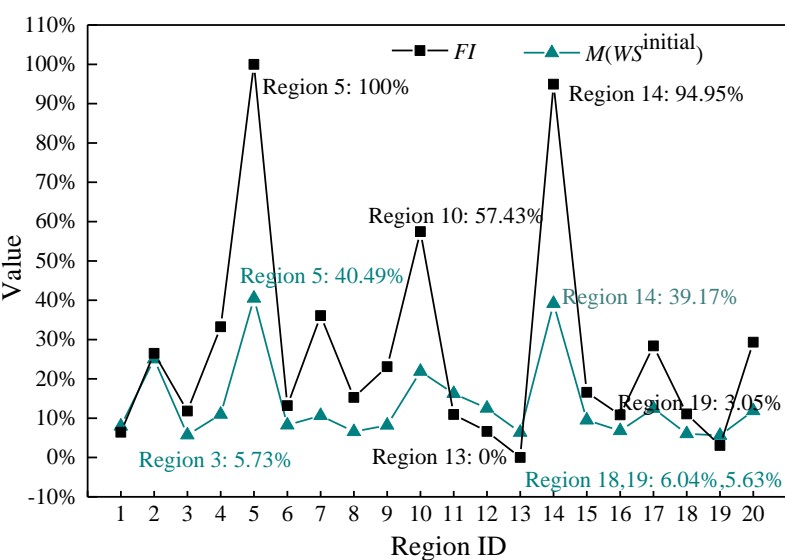

**Figure 9.** The $M(WS^{\text{initial}})$ and *FI* of regions.

Figures 10 and 11 show the results of water shortage caused by disruptions in region 5 and region 14, respectively. *WS* is the water shortage considering the importance of function zone, and $WS^{\text{initial}}$ is the water shortage without considering the importance of function zone. From Figures 10 and 11, it can be seen that $WS^{\text{initial}}$ is different from that of *WS* throughout all regions. In Figure 9, the $WS^{\text{initial}}$ of region 8 was the largest (except for the disrupted region 5), while the *WS* of region 7 is largest considering the urban function zone attributes. Compared with region 8, region 7, in which business and medical zones are located, provides more critical services to the urban area. Hence, the largest *WS* was driven by the water shortage of 74.4% in region 7 although the $WS^{\text{initial}}$ in region 8 (81.23%) was

larger than that in region 7. Moreover, the $WS^{initial}$ of region 19 caused by the disruption in region 5 was relatively large, but the estimated $WS$ was less because there are only a small residential community and parks which are not critical to the urban function. Furthermore, in Figure 11, the $WS^{initial}$ in regions 15, 16 and 17 are the same as each other, i.e., the $WS^{initial}$ of 100% while the estimated $WS$ in region 17 was the largest among them as the largest hospital in the entire district is located in region 17. The $WS^{initial}$ in region 7 was 62.36% but the estimated $WS$ was much larger because there are business and medical zones that provide more critical services to the urban area. These results indicate that the criticality of urban function zones in the WDN needs to be considered to comprehensively evaluate damage-driven $WS$ in a system.

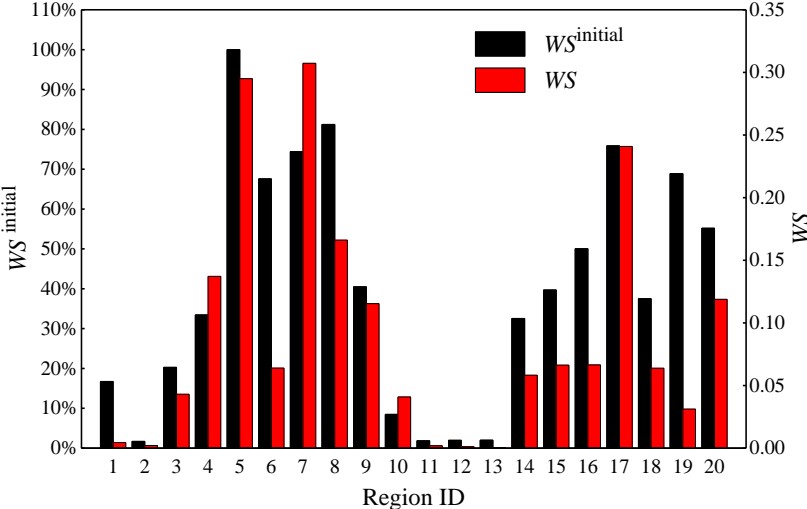

**Figure 10.** The water shortage of regions caused by region 5 disruption.

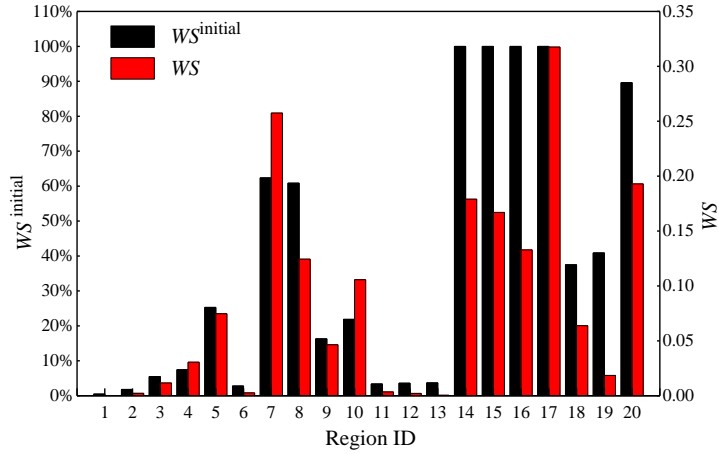

**Figure 11.** The water shortage of regions caused by region 14 disruption.

In the WDN, the smaller the water supply level (structural importance) of a region is, and the more $WS$ (functional importance) in WDN under disruption, then the more critical this region is. In combination with the nodal SI and FI caused by regional damage, the criticality of each region is determined comprehensively. Equation (15) is used to evaluate the critical region. The *SI, FI* and *CI* of regions are shown in Figure 12. From Figure 12, it can be seen that region 5 is the most critical region in the WDN, as the disruption in region 5 caused the largest comprehensive influence on the WDN and urban supplied by the WDN, while the criticality of region 18 was the lowest, and its disruption can lead to the least comprehensive influence on the WDN and urban areas. Furthermore, the *CIs* of regions 1 and 19 were only 0.463 and 0.432, respectively, which are much smaller than the maximum *CI* 0.917 (region 5), although they play the most important role in the topological structure of the WDN. It also

can be seen that the criticality ranking of each region was neither the same as the individual structural importance ranking nor the individual functional importance ranking. It indicates that either of the two metrics is not enough to comprehensively identify the criticality of the regions. As the reinforcement resources for WDNs are usually limited, it is not economically feasible to reinforce the regions in WDNs simultaneously. The more critical regions, the higher priority for reinforcement. For more efficient reinforcement strategy during the pre-disaster preparation phase in a WDN, therefore, it is necessary to comprehensively analyze and identify the critical region in the WDN by the two metrics (i.e., nodal water supply level and functional importance) which takes into account the three measures, the criticality of function zones, nodal supply water level and the water shortage.

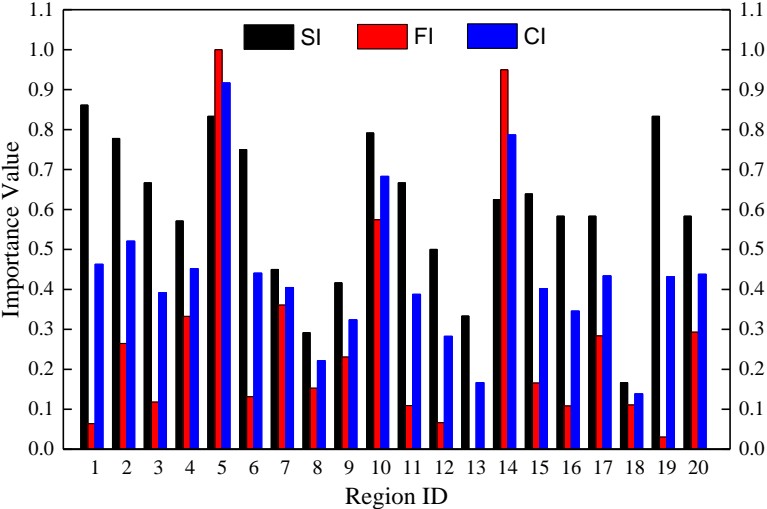

**Figure 12.** Structural importance (SI), functional (FI) and critical importance (CI) of regions.

## 4. Conclusions

Identifying the regions critical to WDN reinforcement is of great assistance for formulating the pre-disaster preparation strategy. When the WDN is damaged by disasters, its topological structure changes and the water flow is redistributed, the pressure of some nodes are lower than the normal service pressure, so some function zones are short of water. Generally, the same water shortage causes greater losses to the more important function zones. Therefore, to identify the critical regions in the WDN, not only the characteristics of WDN itself, but also the importance of the function zones supplied by WDN should be considered. Taking the node failures and pipe breaks as two failure modes, a methodology framework is proposed to identify the critical region in the WDN, which is the basis of the pre-disaster reinforcement strategy. In this study, the proposed methodology framework takes into account the three measures, namely, the criticality of function zones, the nodal water supply level and the water shortage, which consists of three main sections: (1) a method for determine the relative importance of function zones divided into inter-type and intra-type; (2) the nodal water supply level and the water shortage metrics are analyzed from static-equilibrium and pressure-driven hydraulic simulation; and (3) a model for identifying the critical region in the WDN.

The framework was applied to the WDN in Dalian, China, as a case study, and it identified the most critical region in the entire WDN based on three principles: (1) the region in which the nodes have a high water supply level; (2) the water shortage caused by a disruption in the region should be larger; and (3) the importance of urban function zones supplied by the region should be larger. These three principles need to be considered simultaneously to evaluate the critical regions in the WDN comprehensively as no single principle can comprehensively reflect the criticality of the region in the WDN. Scientific reinforcement strategies are required to improve the disaster resistance capacity of the WDN in the pre-disaster stage, leading to reducing the damage and the impacts on the society and economy. The criticality of the region can provide efficient guidance for developing the

pre-disaster reinforcement strategies for water distribution networks. In the limited reinforcement resources, the more critical the region in the WDN, the higher the priority for reinforcement.

In addition to applying the proposed methodology framework to different WDNs to identify the critical region comprehensively, in the search for the more effective guidance for the WDN pre-disaster reinforcement aimed at minimizing the service loss from the preparation resilience, future studies should be about various type of disruptive events. The other failure mode, pipe leakage will be considered in the future studies. Moreover, the valves and pumps that are crucial to system operation normally should be also considered in the methodology framework. A more comprehensive reinforcement strategy considering the security of these facilities is of great guidance for improving the WDN resilience.

**Author Contributions:** The authors provided the following contributions to this paper. Conceptualization, M.Z. and G.L.; methodology, J.Z.; formal analysis, J.Z.; data curation, M.Z. and G.L.; writing—original draft preparation, M.Z. and J.Z.; writing—review and editing, M.Z. and Y.Z.; supervision, M.Z. and Y.Z. All authors have read and agreed to the published version of the manuscript.

**Funding:** This research was funded by the Fundamental Research Funds for the Central Universities, grant number DUT20ZD401 and National Key Research and Development Project, grant number 2018YFD1100405.

**Conflicts of Interest:** The authors declare no conflict of interest.

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
