# Peer review of "A Framework for Identifying the Critical Region in Water Distribution Network for Reinforcement Strategy from Preparation Resilience"

_sustainability, doi:10.3390/su12219247_

Round 1

Reviewer 1 Report

The reviewed text is a new version of the article I reviewed earlier.
In the new version, the authors have significantly expanded all sections of the article.

Comparing my earlier comments with the introduced changes, I can say that most of them authors took into account.

There is still no reference to the risk theory and a comparison of author’s  method with another one, but I estimate that the article can be published in its current form.

Reviewer 2 Report

The authors adequately addressed all the comments. The revised manuscript improved significantly. I will recommend the acceptance of the paper from my side

This manuscript is a resubmission of an earlier submission. The following is a list of the peer review reports and author responses from that submission.

Round 1

Reviewer 1 Report

The authors presented critical factors in the water distribution system highly important for sustainability. The manuscript is scientifically sound and can be accepted by addressing the following comments;

  1. The author should compare the earlier reports regarding this topic.
  2. The author should describe more about critical regions.
  3. The conclusion section should be revised on emphasizing how WDN would be applicable

Author Response

Thank you very much for your valuable comments.We have responded one by one in the attachment.

Reviewer 2 Report

The topic of the article covers the functioning of the critical infrastructure, which has been popular in the scientific and technical community recently. It will certainly be interesting for readers. The form of the article is correct.

Intriouction

The first paragraph is completely devoid of citations, although it contains many previously published statements.

Regarding types of failures - it is worth looking at the following items

Shier D.R. Network reliability and algebraic structures. Oxford: Clarendon Press; 1991.

Kansal M.L., Devi S. An improved algorithm for connectivity analysis of distribution networks. Reliability Engineering and System Safety 2007;  92: 1295–1302

Line 62 - it is worth looking at the possibilities of fractal description and classification of the network, e.g. through the fractal dimension. This allows for a departure from the individual description of each network:

Kowalska B., Kowalski D., Holota E. Fractal-Heuristic Method of Water Quality Sensor Locations in Water Supply Network. Water 2020, 12(3), 832; https://doi.org/10.3390/w12030832

In the literature review, the authors do not refer to the criticality analysis offered as a standard tool in the Bentley-WaterGems program. They also do not refer to risk analysis, although they later apply the methodology of this field of science.

Materials and methods

Fig. 1 Congratulations – it is a very good idea of presentation

Fulfilling the equation 5 requires assigning a sign to the flow direction - proposes to complete the text.

Line 175 and others – the analogical classification you can find in literature – see comment in line 62

Line 183 – citation lack

Fig. 2 – „riendship” ?

Whether the division of the network into areas takes into account the possibility of cutting off each of these areas or is rather related to geographical affiliation, it is worth to clarify.

Can the weighting be automated, e.g. by using the GIS database of water companies regarding the type of recipients?

Results

Line 258 – 20 regions – is it the same that DMA or pressure zones? 

Fig. 4 – what does the red dot mean? Waterr source? Pumping station?

Tab. 1  and lines 268-269 - unclear how the Water supply level was calculated - this should be explained in advance in the Materials and Methods section.

It is a pity that the article does not compare with any other method. The obtained results were not validated in this way. I suggest that the Authors consider the next article in which they will make such a comparison

Author Response

Thank you very much for your valuable comments. We have responded one by one in the attachment.
